# Low Levels of Supplementation for Post-Weaning Girolando Steers on Tropical Pasture During the Dry to Rainy Season Transition

**DOI:** 10.3390/vetsci12040384

**Published:** 2025-04-18

**Authors:** Wbeimar Yamit Sanchez Dueñez, Diana Carolina Cediel-Devia, Osman Ronaldo Aguilar Melgar, Marceliana da Conceição Santos, Sinvaldo Oliveira de Souza, Laize Vieira Santos, Rayce Aparecida Ferreira, Pedro Fernando Caro Aponte, Jeferson Camilo Ortiz Riobo, Fábio Andrade Teixeira, Víctor Gerardo Petro Hernández, Dorgival Morais de Lima Júnior, Robério Rodrigues Silva

**Affiliations:** 1Department of Veterinary Medicine, Campus Universitário Square, Universidade Federal de Viçosa, P H Rolfs Street, Viçosa 36570-900, Minas Gerais, Brazil; wbeimar.duenez@ufv.br (W.Y.S.D.); rayce.ferreira@ufv.br (R.A.F.); pfcaroap@ut.edu.co (P.F.C.A.); 2021m0305@uesb.edu.co (J.C.O.R.); victor.hernandez@ufv.br (V.G.P.H.); 2Department of Animal Science, Universidade Federal de Lavras, Professor Edmir Sá Santos Traffic Circle, Lavras 37203-202, Minas Gerais, Brazil; diana.devia@estudante.ufla.br; 3Department of Rural and Animal Technology, Universidade Estadual do Sudoeste da Bahia, Primavera Square, Primavera, Itapetinga 45700-000, Bahia, Brazil; 94oram@gmail.com (O.R.A.M.); marciazootec@hotmail.com (M.d.C.S.); sinvalsouza79@hotmail.com (S.O.d.S.); santos.laize@yahoo.com.br (L.V.S.); fteixeira@uesb.edu.br (F.A.T.); rrsilva.uesb@hotmail.com (R.R.S.); 4Department of Animal Science, Universidade Federal Rural do Semi-Árido, Mossoró 59625-900, Rio Grande do Norte, Brazil

**Keywords:** dairy beef steers, grazing supplementation, nutrient intake, performance

## Abstract

This study evaluated two levels of concentrate supplementation (1 g/kg and 2 g/kg of body weight, BW) on the intake, digestibility and performance of post-weaning Girolando steers on tropical pastures during the dry to rainy transition season/period. The main problem addressed was how to optimize nutrition in this period when the forage quality decreases, affecting animal growth. The results showed that steers that received 1 g/kg of the supplement had greater forage intake and fiber digestibility, while those that received 2 g/kg consumed more non-fiber carbohydrates. However, the average daily gain and final body weight were similar between the two groups. This study concludes that a supplementation level of 1 g/kg of BW is sufficient to increase nutrient utilization without compromising animal performance, offering a cost-effective solution for beef production in tropical regions. This can help to improve the livestock efficiency, ensuring better use of the available resources.

## 1. Introduction

Most of the worldwide beef production occurs in tropical countries, where the activity is carried out mainly under grazing conditions, whether for environmental or economic reasons [1], thus forming a system with multiple interactions among three major types of resources: basal nutritional resources, supplementary nutritional resources and animal genetic resources [2]. Regarding the basal nutritional resources (i.e., forage), which refers to whatever is present in the production system, in tropical conditions, they tend to go through different physicochemical stages, with great variations in the quantity and quality of their basal structure [3].

In tropical countries, it is necessary to develop tools to overcome these variations in forage nutrients through supplementary nutritional resources that are external to the production system [4,5]. It is important to highlight that the implementation of supplementation levels should be considered only after the maximum potential for use of the forage has been reached, without falling into the substitution effect, as, depending on the level of inclusion of the concentrate in the diet, forage intake is replaced by the energy density of the supplement [6,7].

The genetic resources will be responsible for the transformation and capitalization of the environmental conditions and management of the production system, being maximized by animals adapted to tropical conditions, generating products with economic and environmental efficiency [8,9]. Particularly in the raising of dairy beef steers, there are difficulties in replacing animals for a new production cycle, giving rise to the use of crossbred animals with the capacity to adapt to grazing and representing a solution to overcome inflated costs in livestock [10]. In this context, during the post-weaning phase of beef cattle, higher profitability occurred when using a low supplementation level [11].

The objective of this study was to evaluate the effects of two levels of concentrate supplementation (1 g/kg or 2 g/kg of BW) on the intake, apparent digestibility and performance of Girolando steers on tropical pastures during the post-weaning phase in the dry to rainy season transition.

## 2. Materials and Methods

### 2.1. Locale, Animals and Experimental Design

The experiment was conducted in the municipality of Ribeirão do Largo (15°26′46″ S and 40°44′24″ W), in the southwestern region of the state of Bahia, Brazil, during the dry to rainy season transition. The region features a tropical climate with a dry season of type Aw, according to Köppen–Geiger. Twenty Girolando steers (1/2 Holstein x 1/2 Zebu) were used, with an average initial body weight (iBW) of 151.15 ± 50 kg and 12 months of age. At the beginning of the experimental period, the animals were subjected to ecto- and endoparasite control. The experimental period lasted 275 days, with the initial 14 days dedicated to adapting the animals to the management and diets.

Animals were equally and randomly distributed in a completely randomized design with 2 treatments: SC1, concentrated supplementation, 1 g/kg BW, and SC2, concentrated supplementation, 2 g/kg BW (Table 1). There were ten replicates per treatment. The supplement was consistently provided at 10:00 in uncovered troughs crafted from reused plastic barrels, accessible from both sides (70 linear cm per animal). Diets were formulated following the NRC [12] guidelines to meet the nutritional requirements for animals with a BW of 250 kg, predicting gains of 0.5 kg/day (Table 1).

The animals were allocated to a 14 ha area, comprising 12 paddocks of approximately 1.17 ha each, planted with *Urochloa brizantha* cv. Marandu (Table 2). These paddocks were organized into three modules, each containing four paddocks, equipped with troughs and automatic drinkers. Occupancy within each module lasted for 28 days, with relocations of treatment animals occurring within the module’s paddocks every seven days. This periodic reshuffling aimed to minimize the impact of the paddocks on the steers’ performance. After 28 days, a new module was occupied, and relocations within the module followed the same procedure as previously described.

### 2.2. Forage Evaluation

The forage assessment was carried out every 28 days at both the entrance and exit of the paddocks. To assess the availability of total forage dry matter, the comparative visual yield method [13] was used. The simulated grazing technique was used to estimate the chemical composition of the consumed forage. This technique consists of observing the grazing behavior of animals (measuring the height of the grazed stratum and morphological components) and collecting forage material according to this behavior [14]. To calculate the forage offer (FO; kg DM/100 kg BW/day), it was necessary to determine the dry matter residual biomass (DMRB), which was estimated via the double sampling method [15] and dry matter daily accumulation rate (DAR), as proposed by Campbell [16]. Thus, the FO was calculated according to Prohmann [17]. The estimation of the potentially digestible dry matter (pdDM) of the pasture was performed according to Paulino et al. [18].

### 2.3. Collection, Processing and Analysis of Food Samples

Forage samples from simulated grazing were obtained following Johnson [19]. These samples were utilized to estimate the nutrient intake and apparent digestibility coefficients. Samples of the concentrate supplement were collected in each period using an aliquot, and, at the conclusion of the experiment, a composite of all material was created. The supplement and forage samples were dried in a forced-air oven at 55 °C and ground in a Wiley mill to 1 mm for chemical analyses.

The dry matter (DM), ash, crude protein (CP) and ether extract (EE) content were determined according to the AOAC [20] methodology. Ash- and protein-free neutral detergent fiber (NDFap) was measured as described by Mertens [21]. Non-fibrous carbohydrates were determined, also free of ash and protein (NFCap), by the following equation: NFCap = 100 − ash − CP − EE − NDFap. Because the supplement contained urea, its NFCap content was determined by the following equation: NFCap = 100 − ash − EE − NDFap − (CP − CPu + U), where CPu is the CP in urea and U is the urea content [22]. The total digestible nutrients (TDN) were calculated using the equation TDN% = DCP + DNDFap + DNFC + 2.25 DEE, where DCP is digestible CP; DNDFap is digestible NDFap; DNFC is digestible NFC; and DEE is digestible EE [23].

### 2.4. Intake and Apparent Digestibility

The estimation of fecal excretion was performed by using chromic oxide (Cr_2_O_3_), supplied daily in a single dose of 10 g/animal/d for 12 days. The Cr_2_O_3_ analysis was carried out following the methodology of Detmann et al. [24], using nitroperchloric digestion and with the reading being performed on an atomic absorption spectrophotometer (GBC model, Avanta Sigma, Melbourne, Australia). Subsequently, the daily fecal excretion was calculated according to Smith and Reid [25]. The estimation of the forage dry matter intake was carried out using the undigestible NDF as an indicator (uNDF). From the values of the fecal production and uNDF, we estimated the forage dry matter intake, as described by Detmann et al. [24].

The individual intake of supplement dry matter (DMIs) was estimated with titanium dioxide (TiO_2_) as an indicator, which was offered at 15 g/animal/day, mixed with the concentrated supplement, for 11 days. Chromic oxide was used as an indicator for fecal excretion, being supplied daily at 10:00 AM, as described by Valadares Filho et al. [26]. The titanium concentration and DMIs were estimated according to the methodology described by Detmann et al. [24].

With the values of the forage dry matter intake and supplement dry matter intake, we estimated the total dry matter intake. The digestibility of dry matter and nutrients was determined as described by Silva and Leão [27].

### 2.5. Animal Performance

The BW assessments were preceded by a 12 h feed fasting period and carried out at the beginning of the experimental period and every 28 days thereafter to adjust the amount of the supplement. At the end of the experimental period, the animals were weighed again to evaluate the average daily gain (ADG).

Animal performance was determined by the difference between iBW and the final body weight (fBW), divided by the experimental period of 261 days. Feed conversion (FC) was determined based on the intake and ADG.

### 2.6. Statistical Analysis

Data were analyzed using SAS (Version 9.1; SAS Institute, Inc., Cary, NC, USA), considering a completely randomized design:Yij = μ + Ti + eij
where Yij, observed response; μ, constant; Ti, effect of treatment (SC1 or SC2); and eij, error, NID~(0, σ2).

The data were subjected to the Shapiro–Wilk test and Bartlett test to verify the normality of errors and homogeneity of variances, respectively. The data were thus subjected to an analysis of variance. The significance level adopted was α = 0.05.

## 3. Results

The forage allowance was increased for the transition period between the dry and rainy seasons (Table 3).

The forage dry matter intake (%BW) and neutral detergent fiber intake (NDF) were higher (*p* < 0.05) for steers supplemented with a level of 1 g/kg of BW; however, the supplement concentrate intake (kg/day) and non-fibrous carbohydrate (NFC) intake were higher (*p* < 0.05) for steers consuming 2 g/kg BW of the concentrated supplement (Table 4).

The dry matter and organic matter digestibility were not influenced by the level of supplementation. The NDF and NFC digestibility were higher (*p* < 0.05) for steers consuming 1 g/kg BW and 2 g/kg BW of the concentrated supplement, respectively.

There was no effect (*p* > 0.05) of supplementation on the BW at slaughter, ADG and FC (Table 5).

## 4. Discussion

Promoting an increase in forage intake is the main objective when using supplements in pasture-based production systems [28]. Protein supplementation may provide an increase in forage intake when the TDN/CP ratio of the pasture is greater than 70.0 g/kg [29]. However, in this study, the value of this ratio was 59.7 g/kg, which explains the lack of increase in forage intake with higher levels of supplementation [30]. The absence of differences in total dry matter intake may be associated with the high forage allowance [31].

In the present study, the forage dry matter intake was similar between the supplementation strategies, but, when considering the BW, it was higher for SC1 animals. It was possible to notice a replacement in forage intake (kg/day) by supplement intake in SC2 animals [5]; comparable results were reported by Moraes et al. [32].

An increase in apNDF intake can be assumed if the forage dry matter intake differs, since protein supplementation could lead to an increase in forage intake [33]. However, Paula et al. [34] found that the NDF intake (kg/day) was reduced with increasing levels of supplementation.

The CP intake in this study was sufficient to meet the CP intake requirements recommended for steers [35]. Moreover, the percentage of CP in the total diet was 12.8%, which is sufficient to guarantee the development of fibrolytic bacteria in the ruminal environment [36]. Furthermore, CP levels close to 11% maximize nutrient intake and digestibility under forage-based feeding conditions [37]. In the present study, the NDF digestibility coefficient was higher in SC1 animals. Since the SC2 treatment had an increase in the forage–concentrate ratio due to the greater supplement intake, this may have increased the intake of carbohydrates with rapid ruminal fermentation, changing the rumen pH and affecting fibrolytic microorganisms [38,39]. The NFC digestibility coefficient increased with supplementation and with increasing supplement intake, due to the greater amount of grain supplied to the animals via the supplement, which led to greater NFC intake [40].

The productive performance of animals raised in pasture systems is related to the quantity and quality of forage available for grazing, with intake being the main determinant of animal performance [31,41]. The similar performance between the two treatments in this study is explained by the quality of the dry matter and green dry matter, with values above those recommended by Costa et al. [42] as necessary to positively influence selectivity and offer greater nutritional supplies to animals. It is important to highlight that, under the conditions of this study, it was more beneficial to offer the lowest level of supplementation, as this was sufficient to achieve the best results. Sales et al. [43] reported that higher supplementation levels improve the performance of steers finished on tropical pastures.

## 5. Conclusions

Increasing the supplementation level from 1 g/kg BW to 2 g/kg BW does not improve the performance of steers during the post-weaning phase, but the 1 g/kg BW level increases pasture use by steers. Therefore, it is advisable to provide concentrate supplements at a level equivalent to 1 g/kg of BW for steers on tropical pastures during the dry to rainy season transition.

## Figures and Tables

**Table 1 vetsci-12-00384-t001:** Composition (g/kg) of supplements, forage availability (kg/ha) and chemical composition of the forage and the supplement.

	*U. brizantha* cv. Marandu ^1^	Supplement
Ground sorghum grain	-	560
Soybean meal	-	200
Urea	-	150
Mineral salt ^2^	-	90
Chemical composition (g/kg)		
Dry matter (g/kg as fed)	282.1	893.0
Ash (g/kg of dry matter)	97.6	107.0
Crude protein (g/kg of dry matter)	95.3	583.5
Ether extract (g/kg of dry matter)	17.5	36.6
apNDF ^3^ (g/kg of dry matter)	652.5	160.0
NFC ^4^ (g/kg of dry matter)	139.5	243.7
TDN ^5^ (g/kg of dry matter)	569.3	586.5

^1^ Simulated grazing; ^2^ guaranteed levels: calcium 175 g; phosphorus 60 g; sodium 107 g; sulfur 12 g; magnesium 5000 mg; cobalt 107 mg; copper 1300 mg; iodine 70 mg; manganese 1000 mg; selenium 18 mg; zinc 4000 mg; iron 1400 mg; fluoride (maximum) 600 mg; ^3^ neutral detergent fiber corrected for ash and protein; ^4^ non-fibrous carbohydrates; ^5^ estimated total digestible nutrients.

**Table 2 vetsci-12-00384-t002:** Chemical composition of the total diet consumed by the animals during the experimental period.

Component (g/kg)	Supplementation Level	
	1 g/kg BW ^1^	2 g/kg BW ^1^
Dry matter (g/kg as fed)	309.0	338.2
Ash (g/kg of dry matter)	98.0	98.5
Crude protein (g/kg of dry matter)	116.6	139.9
Ether extract (g/kg of dry matter)	18.3	19.3
apNDF ^2^ (g/kg of dry matter)	630.3	606.7
NFC ^3^ (g/kg of dry matter)	143.6	148.6
TDN ^4^ (g/kg of dry matter)	570.1	570.9
**Forage to concentrate ratio (%)**		
Forage	95.58	90.80
Concentrate	4.42	9.20

^1^ Body weight; ^2^ neutral detergent fiber corrected for ash and protein; ^3^ non-fibrous carbohydrates; ^4^ estimated total digestible nutrients.

**Table 3 vetsci-12-00384-t003:** Characteristics of the pasture (*Urochloa brizantha* cv. Marandu) during the dry to rainy season transition.

Variable	Value (kg/ha)
Dry matter total availability (DMTA)	3364
Green dry matter availability (GDMA)	2527
Potentially digestible DM (pdDM)	2737
Forage allowance DM (%) (FA DM)	16.56
Leaf-to-stem ratio	1.11
Leaf	1334
Stem	1193
Dead material	837.2

**Table 4 vetsci-12-00384-t004:** Nutrient intake and apparent digestibility of Girolando steers supplemented with low levels of concentrate while grazing on tropical pasture during the dry to rainy season transition.

	Supplementation Level		s.e. ^2^	*p*-Value ^3^
	1 g/kg BW ^1^	2 g/kg BW ^1^		
**Intake (kg/d)**				
Total DM ^4^	4.76	4.46	0.18	0.255
Total DM (%BW ^1^)	2.25	2.08	0.05	0.092
Forage DM	4.55	4.05	0.17	0.060
Forage DM (%BW ^1^)	2.15	1.89	0.05	0.009
Supplement DM	0.21	0.41	0.02	<0.001
Crude protein	0.54	0.58	0.02	0.344
Ether extract	0.08	0.07	0.003	0.801
apNDF ^5^	2.82	2.54	0.11	0.085
apNDF (%BW ^1^)	1.31	1.18	0.03	0.013
NFC ^6^	0.75	0.78	0.03	0.590
TDN ^7^	2.54	2.38	0.10	0.277
**Apparent digestibility (%)**				
DM ^4^	58.68	58.73	0.48	0.935
Crude protein	53.56	56.60	1.31	0.121
Ether extract	66.67	66.73	2.22	0.983
apNDF ^5^	57.59	55.38	0.63	0.025
NFC ^6^	66.24	71.00	1.26	0.013
TDN ^7^	53.35	53.36	0.57	0.954

^1^ Body weight; ^2^ standard deviation of the mean; ^3^ significant at the 0.05 probability level; ^4^ dry matter; ^5^ neutral detergent fiber corrected for ash and protein; ^6^ non-fibrous carbohydrates; ^7^ total digestible nutrients.

**Table 5 vetsci-12-00384-t005:** Performance of Girolando steers supplemented with low levels of concentrate while grazing on tropical pasture during the dry to rainy season transition.

	Supplementation Level		s.e. ^2^	*p*-Value ^3^
	1 g/kg BW ^1^	2 g/kg BW ^1^		
Initial BW ^1^ (kg)	149.7	148.0	4.32	0.97
Final BW ^1^ (kg)	297.4	298.4	5.00	0.94
Average daily gain (kg/d)	0.57	0.58	0.0005	0.75
Feed conversion (kg/kg)	8.86	7.45	0.99	0.07

^1^ Body weight; ^2^ standard deviation of the mean; ^3^ significant at the 0.05 probability level.

## Data Availability

None of the data were deposited in an official repository.

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
