# Peer review of "Low Levels of Supplementation for Post-Weaning Girolando Steers on Tropical Pasture During the Dry to Rainy Season Transition"

_vetsci, 2025, doi:10.3390/vetsci12040384_

Round 1
Reviewer 1 Report
Comments and Suggestions for Authors
The study addresses an important issue in tropical livestock production—optimizing concentrate supplementation strategies for Girolando steers. However, significant shortcomings compromise the scientific rigor and credibility of the findings.
The reported crude protein (CP) level in the concentrate, at 583.3 g/kg, is exceptionally high and unrealistic for tropical grazing systems. This figure deviates substantially from standard nutritional formulations, raising serious concerns about data accuracy. Such a discrepancy casts doubt on whether the supplementation levels used were appropriate for evaluating performance outcomes in Girolando steers. So the validity of the findings is questionable.
Additionally, the standard deviation (SD) for feed conversion is alarmingly greater than the mean in both supplementation groups (9.99 vs. 8.86 and 7.45, respectively). This indicates substantial variability in the feed conversion ratio (FCR), suggesting the presence of outliers, a skewed distribution, or calculation errors. Such extreme variability is atypical and significantly undermines the reliability and interpretability of the results. The authors must revise their statistical approach accordingly.
Finally, the study lacks statistical analysis of initial body weight (BW), a critical factor for ensuring the comparability of treatment groups. Without such analysis, it is unclear whether observed effects on performance can be attributed to the supplementation strategy or pre-existing differences between groups.
Author Response
Review 1.
The reported crude protein (CP) level in the concentrate, at 583.3 g/kg, is exceptionally high and unrealistic for tropical grazing systems. This figure deviates substantially from standard nutritional formulations, raising serious concerns about data accuracy. Such a discrepancy casts doubt on whether the supplementation levels used were appropriate for evaluating performance outcomes in Girolando steers. So the validity of the findings is questionable.
Answer: Dear Reviewer, we disagree with your statement. The 583.3 g/kg CP content is from the supplement, not from the pasture or total diet. The animals received either 1 g/kg CP or 2 g/kg CP, resulting in an intake of 112 g/kg CP and 130 g/kg CP in the total diet. Please refer to our intake table.
Additionally, the standard deviation (SD) for feed conversion is alarmingly greater than the mean in both supplementation groups (9.99 vs. 8.86 and 7.45, respectively). This indicates substantial variability in the feed conversion ratio (FCR), suggesting the presence of outliers, a skewed distribution, or calculation errors. Such extreme variability is atypical and significantly undermines the reliability and interpretability of the results. The authors must revise their statistical approach accordingly.
Answer: Dear reviewer, we agree with you. We made a mistake. The correct value is not 9.99 but 0.99. We have made the corrections.
Finally, the study lacks statistical analysis of initial body weight (BW), a critical factor for ensuring the comparability of treatment groups. Without such analysis, it is unclear whether observed effects on performance can be attributed to the supplementation strategy or pre-existing differences between groups.
Answer: Dear reviewer, we agree with you. Statistics have been added for this variable.

Reviewer 2 Report
Comments and Suggestions for Authors
This short communication assesses the impact of low-level supplementation for post-weaning Girolando steers on tropical pastures during the dry to rainy season transition. This study could provide viable feeding strategies to improve beef economics in the tropics. However, there are still some problems and shortcomings in the paper that should not be ignored. Here are some of my comments and suggestions.
1. Line 23: The two groups had no significant change in average daily gain or final body weight, so what about their feed intake?
2. Line 55-59: What occurs during the transition from the dry to the rainy season in the absence of supplemental feeding? The authors must present this information clearly and concisely. In the absence of a critical situation, the establishment of a control group for comparative analysis would be advisable.
3. Line 82-84: How are these two levels of supplementation determined? The conclusion of the article recommends a supplementation level of 1 g/kg at relatively low levels and low cost, so has a supplementation level of less than 1 g/kg been considered? The experimental design lacks rigor. You should have set up a few more horizontal gradients to explore the results of supplemental feeding.
4. Line 229-230: “Increasing the supplementation level from 1 g/kg to 2 g/kg increases the use of nutrients but does not improve the performance of steers during post-weaning phase.” Is this an appropriate formulation of the sentence? In line 23-25: Your study concluded that supplementation with 1 g / kg of body weight increased the nutrient utilization of Girolando steers.
Author Response
Revisor 2.
- Line 23: The two groups had no significant change in average daily gain or final body weight, so what about their feed intake?
Answer: Dear Reviewer, we did not actually observe any effect on performance, even with the differences observed in nutrient intake and digestibility. Please consider that increased intake and digestibility may not directly influence animal performance.
- Line 55-59: What occurs during the transition from the dry to the rainy season in the absence of supplemental feeding? The authors must present this information clearly and concisely. In the absence of a critical situation, the establishment of a control group for comparative analysis would be advisable.
Answer: Dear Reviewer, please consider that emphasis has been placed on the transition period to mark the changes that occur in pasture characteristics.
- Line 82-84: How are these two levels of supplementation determined? The conclusion of the article recommends a supplementation level of 1 g/kg at relatively low levels and low cost, so has a supplementation level of less than 1 g/kg been considered? The experimental design lacks rigor. You should have set up a few more horizontal gradients to explore the results of supplemental feeding.
Answer: Dear Reviewer, we ask that you consider the supplementation levels as low. Our group has tested these levels in other published manuscripts (https://doi.org/10.4025/actascianimsci.v46i1.69224; https://doi.org/10.21071/az.v59i228.4710; https://doi.org/10.1007/s11250-018-1574-y; https://doi.org/10.1007/s11250-019-01824-2 )
- Line 229-230: “Increasing the supplementation level from 1 g/kg to 2 g/kg increases the use of nutrients but does not improve the performance of steers during post-weaning phase.” Is this an appropriate formulation of the sentence? In line 23-25: Your study concluded that supplementation with 1 g / kg of body weight increased the nutrient utilization of Girolando steers.
Answer: Dear Reviewer, we agree with you. I made a small change to the conclusion.

Reviewer 3 Report
Comments and Suggestions for Authors
Dear Author,
Please to read this paper which you can found the following url side.
I think the more than 10 years ago published manuscript interesting for your research
https://doi.org/10.1590/S1516-35982011000400027
Author Response
Revisor 3.
Please to read this paper which you can found the following url side. I think the more than 10 years ago published manuscript interesting for your research. https://doi.org/10.1590/S1516-35982011000400027
Answer: Dear Reviewer, we agree with you. We have inserted the reference in the text and in the list of references.

Round 2
Reviewer 1 Report
Comments and Suggestions for Authors
The author must justify the use of this specific concentrate with a crude protein content of 58.35%. Without such justification, it is difficult to assess the relevance and importance of its incorporation rate in the experimental diet. For example add reference of other study used similar CP content in these animals.
Add the reference of calculation: Non-fibrous carbohydrates and Total digestible nutrients
In the discussion section, when comparing intake results with other studies, it is crucial to specify the basis of comparison, intake expressed in terms of metabolic weight or total ingested quantity.
When using abbreviations, define them at their first occurrence and use them consistently throughout the manuscript. For example, Crude protein (CP) in L 127 and 209; Body weight (BW) in L 78, 88, 160, 161, 189, 233 … This approach should be verified across the entire article.
CNF in 182 and d in table 4 are not defined
Table 1 Composition (g/kg) in dry matter basis ? of other
Divergences of result such as in Table 4: 1g/kg BW + Forage DM (%BW) 2.15 = 2.25 ? however the result of total DM (%BW) 2.21 !!! verified all document
Lines 210–212: The statement “Besides, the percentage of CP on the total diet was 12.8%, good enough to guarantee the development of fibrolytic bacteria in the ruminal environment” is unclear. Does this percentage refer to diet 1 or diet 2?
Conclusion: The conclusion that increasing the level of concentrate improves digestibility is inaccurate. The results clearly show that higher concentrate levels reduce NDF digestibility while increasing NFC digestibility. This section should be corrected to accurately represent the findings.
Correct 1 g/kg to 2 g/kg in conclusion by 1 g/kg BW to 2 g/kg BW
Author Response
(Reviewer 1):
The author must justify the use of this specific concentrate with a crude protein content of 58.35%. Without such justification, it is difficult to assess the relevance and importance of its incorporation rate in the experimental diet. For example add reference of other study used similar CP content in these animals.
Answer: Dear Reviewer, we agree with you, please see: https://doi.org/10.21071/az.v59i228.4710 ; https://doi.org/10.1007/s11250-018-1574-y ; https://doi.org/10.1007/s11250-019-01824-2 )
Add the reference of calculation: Non-fibrous carbohydrates and Total digestible nutrients
Answer: Dear Reviewer, we agree with you, we have inserted it in the text.
In the discussion section, when comparing intake results with other studies, it is crucial to specify the basis of comparison, intake expressed in terms of metabolic weight or total ingested quantity.
Answer: Dear Reviewer, we agree with you. But please specify better where we will use the units suggested in the discussion.
When using abbreviations, define them at their first occurrence and use them consistently throughout the manuscript. For example, Crude protein (CP) in L 127 and 209; Body weight (BW) in L 78, 88, 160, 161, 189, 233 … This approach should be verified across the entire article.
Answer: Dear reviewer, please note that this identification of acronyms throughout the text helps in the general understanding of the text.
CNF in 182 and d in table 4 are not defined
Answer: Dear reviewer, we made the corrections.
Table 1 Composition (g/kg) in dry matter basis ? of other
Answer: Dear reviewer, we made the corrections. Dear reviewer, this value is an average between the CP values ​​of the total diet of the two treatments.
Divergences of result such as in Table 4: 1g/kg BW + Forage DM (%BW) 2.15 = 2.25 ? however the result of total DM (%BW) 2.21 !!! verified all document
Answer: Dear reviewer, we made the corrections.
Lines 210–212: The statement “Besides, the percentage of CP on the total diet was 12.8%, good enough to guarantee the development of fibrolytic bacteria in the ruminal environment” is unclear. Does this percentage refer to diet 1 or diet 2?
Answer: Dear reviewer, this value is an average between the CP values ​​of the total diet of the two treatments.
Conclusion: The conclusion that increasing the level of concentrate improves digestibility is inaccurate. The results clearly show that higher concentrate levels reduce NDF digestibility while increasing NFC digestibility. This section should be corrected to accurately represent the findings.
Answer: Dear reviewer, we made the corrections.

Reviewer 3 Report
Comments and Suggestions for Authors
Thnak you for revised manuscript. No further suggestions.
Author Response
Dear Reviewer, thank you very much for your suggestions and improvements to our manuscript.
Round 3
Reviewer 1 Report
Comments and Suggestions for Authors
In the discussion section, when comparing intake results with other studies, it is crucial to specify the basis of comparison, such as intake expressed in terms of metabolic weight or total ingested quantity.
- Line 203: Comparable results were reported by Moraes et al. [32].
- Line 207: Paula et al. [34] found that NDF intake decreased with increasing levels of supplementation.
When using abbreviations, ensure they are defined first after used in all the manuscript. For example, " body weight" should be used as BW (Lines 233, 78, 88, 160, 161, 189). Similar average daily gain, dry matter intake, and crude protein …
Table 1: Precise chemical composition (g/kg) should be based on dry matter or another appropriate basis.
Table 2: Dry matter content should be specified in terms of fresh matter.
Author Response
(Reviewer 1):
In the discussion section, when comparing intake results with other studies, it is crucial to specify the basis of comparison, such as intake expressed in terms of metabolic weight or total ingested quantity.
Line 203: Comparable results were reported by Moraes et al. [32].
Answer: Dear Reviewer, we agree with you, we have inserted it in the text.
Line 207: Paula et al. [34] found that NDF intake decreased with increasing levels of supplementation.
Answer: Dear Reviewer, we agree with you, we have inserted it in the text.
When using abbreviations, ensure they are defined first after used in all the manuscript. For example, " body weight" should be used as BW (Lines 233, 78, 88, 160, 161, 189). Similar average daily gain, dry matter intake, and crude protein …
Answer: Dear Reviewer, we agree with you, we have inserted it in the text.
Table 1: Precise chemical composition (g/kg) should be based on dry matter or another appropriate basis.
Answer: Dear Reviewer, we agree with you, we have inserted it in the table.
Table 2: Dry matter content should be specified in terms of fresh matter..
Answer: Dear Reviewer, we agree with you, we have inserted it in the table.
